# Musculoskeletal and Neuropathic Pain in COVID-19

**DOI:** 10.3390/diagnostics14030332

**Published:** 2024-02-04

**Authors:** Christopher M. Lam, Miles Sanderson, Dan T. Vu, Dawood Sayed, Usman Latif, Andrea L. Chadwick, Peter Staats, Abigail York, Gabriella Smith, Vivek Velagapudi, Talal W. Khan

**Affiliations:** 1Department of Anesthesiology, Pain, and Perioperative Medicine, University of Kansas Medical Center, Kansas City, KS 66160, USA; clam2@kumc.edu (C.M.L.); msanderson2@kumc.edu (M.S.); danvu@kumc.edu (D.T.V.); dsayed@kumc.edu (D.S.); ulatif@kumc.edu (U.L.); anicol@kumc.edu (A.L.C.); ayork2@kumc.edu (A.Y.); 2National Spine and Pain Centers, Frederick, MD 21702, USA; peterstaats@hotmail.com; 3School of Medicine, University of Kansas Medical Center, Kansas City, KS 66160, USA; gsmith16@kumc.edu (G.S.); vvelagapudi@kumc.edu (V.V.)

**Keywords:** musculoskeletal pain, neuropathic pain, COVID-19, pain mechanisms, diagnostic challenges, new techniques, multidisciplinary pain management, epidemiology, pandemic and impact on pain, disability

## Abstract

Chronic pain constitutes a significant disease burden globally and accounts for a substantial portion of healthcare spending. The COVID-19 pandemic contributed to an increase in this burden as patients presented with musculoskeletal or neuropathic pain after contracting COVID-19 or had their chronic pain symptoms exacerbated by the virus. This extensive literature review analyzes the epidemiology of pain pre-pandemic, the costs associated with the COVID-19 pandemic, the impact of the virus on the body, mechanisms of pain, management of chronic pain post-pandemic, and potential treatment options available for people living with chronic pain who have had or are currently infected with COVID-19.

## 1. Introduction

Chronic pain affects both the physical and monetary health of the global population. The impact of COVID-19 on this already strained population has exacerbated these effects, necessitating research into the ways of managing chronic pain symptoms. COVID-19 has led to many challenges in healthcare including an increase in the number of patients who present with COVID-19-related chronic pain as well as the potential for increased pain in patients who contracted COVID-19 and were already living with preexisting chronic pain. Chronic pain associated with COVID-19 infection is postulated to be related to significant levels of systemic inflammation that are caused by an individual’s immune response to the infection, as well as other neuropathic pain mechanisms. A coordinated multidisciplinary approach that promotes physical and psychological interventions is most effective in treating chronic pain. Early evidence suggests a role for novel techniques such as stellate ganglion blocks and vagus nerve stimulation may help to treat symptoms of Long COVID. Additional investigation into pharmacological, non-pharmacological, vagus nerve stimulation, and other various therapies are necessary to understand and treat pain associated with COVID-19.

## 2. Epidemiology of Musculoskeletal and Neuropathic Pain Pre-Pandemic

Defined as “pain in one or more anatomical regions persisting or recurring for longer than 3 months and is associated with significant emotional distress or functional disability… and that cannot be better accounted for by another chronic pain condition”, chronic pain constitutes a significant disease burden and accounts for a sizable proportion of healthcare spending [1]. An analysis of total US healthcare spending in 2016 found that low back and neck pain constitute the highest amount of health care spending at $134.5 billion in costs. Meanwhile several other musculoskeletal disorders account for the second highest cost at $129.8 billion [2]. Musculoskeletal pain can be primarily from the bone, joint, muscle, or related soft tissue themselves. Secondarily, musculoskeletal pain can be defined as part of a disease process that directly affects these areas [1,3].

Cimmino et al. note that the approximate prevalence of musculoskeletal pain is 30% with a range of 13.5–47% [4]. Notable risk factors for development of chronic musculoskeletal pain include age up to 65 years, low education, social isolation, female sex, low family income, depression, anxiety, sleep disorders, non-Caucasian, and history of manual work [4]. Among these risk factors, social disadvantage was found to be associated with worse physical and behavioral health in patients with chronic musculoskeletal pain compared to race and sex [5]. Medical comorbidities and number of painful joints were found to be prognostic factors for elderly patients with musculoskeletal pain [6]. Further, higher number of painful joints may be associated with higher healthcare visits and overall healthcare costs [7]. The degree of disability as well as increase in musculoskeletal pain has been noted to be associated with several physical determinants including increased body mass index (BMI), increased body fat, and sedentary behavior [8,9,10,11].

Kinesiophobia has been implicated as a risk factor for persistent musculoskeletal pain [12,13]. Additional psychosocial aspects of work including job demands, control, support, satisfaction, imbalance between effort with reward, and monotony of occupational tasks have been shown to be associated with musculoskeletal pain [14]. A Dutch study developed a measurement of musculoskeletal health climate evaluating the perceived management priority of musculoskeletal health assessed with work group pain acceptance. Their study showed that there was a link between poor musculoskeletal health climate with increased musculoskeletal pain sites and increased sick absence [15]. With increased ergonomic workplace exposures, there is an increased incidence of musculoskeletal pain [16]. As expected, the incidence of development of musculoskeletal pain increases with duration of occupation and stabilizes with retirement [17]. These factors illustrate the complexity of musculoskeletal pain as physical determinants associated with activity or inactivity as well as psychologic factors can predispose or increase the risk for development of chronic musculoskeletal pain.

Neuropathic pain is defined as pain caused by a lesion or disease of the somatosensory nervous system. It is further classified as peripheral or central in etiology [1]. This newer definition was changed from prior where neuropathic pain was defined as “pain initiated or caused by a primary lesion, dysfunction, or transitory perturbation of the peripheral or central nervous system” [18]. It is often associated with abnormal painful sensations (paresthesias or dysesthesias) as well as neurologic deficits (sensory, cognitive, or motor) depending on the etiology and lesion location [19].

Many disease processes may cause neuropathic pain including diabetes, infection, surgery, primary neurologic disease, trauma, stroke, and cancer [20]. The mechanism of injury is complex and varied; however, it is often attributed to the plasticity of the nervous system itself. Molecular mediators implicated in the development of neuropathic pain include ion channels, immune cells, mitochondrial dysfunction, and glial cells [18]. For example, it has been seen that peripheral nerve injuries cause upregulation of a ligand which activates natural killer cell receptors in the DRG that results in degeneration of the injured nerve by the natural killer cells [18]. Similarly, it is hypothesized that the damage to the somatosensory nervous system in spinal cord injury results in neuropathic pain [18]. Certain genetic factors including mutations to sodium voltage gated channels (Na_v_1.7, Na_v_1.8, Na_v_1.9) have been implicated in certain neuropathic pain conditions including erythromelalgia, diabetic polyneuropathy, and familial episodic pain syndrome [21,22,23]. The prevalence of neuropathic pain is difficult to quantify due the variability of assessment measures, variability of development of neuropathic pain secondary to disease processes, patient response to testing, and the subjective nature of testing for neuropathic pain [24]. As such, it is difficult to obtain a true objective quantifiable incidence of neuropathic pain as well as its implications on society, healthcare, the economy, and patient costs. Despite this difficulty, a general population systemic review of neuropathic pain has quoted that the prevalence of neuropathic pain is between 6.9% to 10.0% [25]. Due to the high prevalence of both pain conditions, musculoskeletal and neuropathic pain already have a far-reaching effect on many individuals at some point in their lives prior to COVID-19.

## 3. The Impact of COVID-19 on Healthcare

The 2019 outbreak of the Severe Acute Respiratory Syndrome Coronavirus 2 (SARS-CoV-2) virus, renamed as COVID-19 by the World Health Organization, generated an international public health emergency with long lasting implications [26]. From masking requirements, vaccination recommendations, social distancing, and global closures to decrease the risk of transmission, multiple fields were impacted, particularly healthcare. In the preliminary stages of the outbreak, there was an increase in critically ill patient volumes. This resulted in a higher consumption of medical equipment (including ventilators, negative pressure rooms, personal protective equipment), increased healthcare staffing demand, increased hours worked by healthcare workers, and increased intensive care unit (ICU) utilization [27]. These experiences have identified deficiencies in many healthcare processes that provide valuable lessons in disaster responses for future events. Serving as a guide, past transgressions may highlight areas to optimize in the future to better cope with eventual high resource demanding crises.

With uncertainty regarding treatment and transmission, healthcare visits were initially triaged to handling those directly impacted by COVID-19 and urgent/emergent situations. This resulted in many elective procedures and surgeries being cancelled amounting to a loss of $202.6 billion dollars in revenue ($50.7 billion per month). Contemporaneously, there was an increase healthcare cost associated with the treatment of those sick with COVID-19 due to utilization of critical resources during this period [27,28]. The federal CARES Act subsequently provided some financial aid by supporting US hospitals through emergency funding, delaying cuts to Medicaid disproportionate share hospital payments, temporary elimination of the Medicare sequester, and Medicare diagnostic-related group add-on payments for treating COVID-19 patients which provided short term relief. However, the effects of the pandemic persisted [28,29]. Policies enacted allowed for increased infection control measures, increased testing, increased hiring, and increased technologic access for various health care entities including hospitals, nursing homes, and long-term care facilities to help protect those at increased risk of contracting COVID [29].

An analysis of US health care resource use (HRU) of patients six months after contracting COVID-19 compared to individuals without a history of index COVID-19 infection showed a statistically significant increased cost at month one (associated with the inpatient hospital cost mean difference of $3706 for commercial and $10,595 for Medicare insurance) after index infection. This increased cost persisted to five months in those infected with COVID-19 [29]. These findings were similar to another retrospective analysis performed with the Premier Healthcare COVID-19 database. In this study, the median inpatient cost to hospitals were quoted as $12,046 and median hospital charges were $43,986. In this cohort, 21.9% of patients were admitted to the ICU and overall inpatient mortality was 13.6%. Patients admitted to the ICU with invasive mechanical ventilation had a 53.8% mortality [30].

Demands placed on healthcare workers and associated fatigue resulted in significant staffing shortages particularly in nursing homes [31,32,33]. Due to increased staffing needs, compensation for emergency staffing increased resulting in a healthcare provider migration from low income to high income countries [34]. With the epidemiologic variabilities of disease prevalence, it has been hypothesized that epidemiologic considerations in designing workforce scheduling may decrease the burden of decreased staffing [35].

An incidental benefit from the pandemic, aside from improved insight on needs from materials and staffing deficiencies in global healthcare systems, may be the advent and utilization of telemedicine [36]. Initially used to prevent disease transmission while providing a means to continue to provide care, the use of telehealth allowed an alternative avenue to provide patient care [37]. This eventually, in combination to other changes to resource utilization, helped eased the patient care burden by allowing for a gradual return to status quo in seeing patients for non-urgent or emergent care. Given the success and rapid adoption of telemedicine, it has been speculated that it may be utilized continually in the future due to its benefits of time conservation, resource conservation, safety improvement, and provided care access for patients who may live in resource strapped locations with poor access to care [38]. Initially utilized as a means to increase patient access during COVID-19, telemedicine is being adopted to increase overall patient access. 

## 4. COVID-19 and Its Impact on Musculoskeletal and Neuropathic Pain

The COVID-19 pandemic negatively impacted patients with musculoskeletal and neuropathic pain. Prior to COVID-19, musculoskeletal and neuropathic pain had a substantial prevalence, affecting up to 47% and 10% of the general population [4,25]. Subsequently, after contracting COVID-19, patients started developing further musculoskeletal pain and neuropathic pain thought to be attributed to the infection itself. Long COVID is defined as a multi system condition comprising severe symptoms following a SARS-CoV-2 infection, affecting at least 10% of all patients who contracted the infection [39]. The exact mechanism of action is unknown, but it is implicated with post infective immune dysregulation, microbiota disruption, autoimmunity, dysfunctional neurologic signaling and possible reactivation of previously dormant viruses including EBV and HHV-6 [39,40]. It is associated with all ages and all COVID disease severities with the highest prevalence in patients between ages 36 and 50 with mild disease presentation initially [41]. Included in the systems affected by Long COVID are cardiovascular, thrombotic, cerebrovascular, endocrinologic, dysautonomic, and musculoskeletal disease. Patients with musculoskeletal manifestations of Long COVID often present with posexertional malaise, fatigue not alleviated by rest, and cognitive impairment (see Figure 1).

It has been estimated that about 10% of patients infected with COVID-19 will develop post-COVID musculoskeletal pain with an overall prevalence of 5.65–18.15% and 4.6% to 12.1% of post-COVID myalgia and joint pain, respectively [41,42]. Animal models of COVID-19 infections have found to develop significant post infective bone loss thought be associated with a post infectious proinflammatory response that may be implicated as a possible cause for post infective musculoskeletal pain [43]. These findings have been associated with imaging changes including muscle edema with necrosis as well as atrophy on magnetic resonance imaging. MR neurography have also found nerve enlargement with signal hyperintensity and loss of fascicular architecture with possible muscle denervation [44].

With restricted healthcare access during the pandemic, delayed musculoskeletal care has increased and resulted in unfavorable musculoskeletal and neuropathic pain-related outcomes [43]. The overall effect is mixed as decreased workplace exposures due to working from home and remote learning may result in less musculoskeletal pain in some while the decreased physical activity accompanied with the utilization of electronic devices will increase shoulder and neck pain in others [45,46]. Telehealth/telemedicine allowed for a greater number of patients to receive physical therapy which resulted in improved musculoskeletal health [47].

Neuropathic pain has also been noted to increase after COVID-19 infection. The exact pathophysiology and mechanism are unknown. Patients present with conditions such as post herpetic neuralgia, trigeminal neuralgia, and brachial plexopathy [48]. It has been suggested that the mechanism of action may be directly related to the COVID infection resulting in a cytokine storm and the proinflammatory effects on the affected nerves or secondarily through reactivation of dormant viruses [39,49,50]. It has been reported that 20% of patients with post-COVID pain fulfilled the criteria of neuropathic pain symptoms through the Self-Report Leeds Assessment of Neuropathic Symptoms while 12% of patients noted having pain with neuropathic features by the PainDETECT questionnaire [51,52].

## 5. Emerging Perspectives on Underlying Mechanisms in Chronic Pain in COVID-19

Understanding the pathophysiologic mechanisms behind chronic pain related to COVID-19 infection is important in identifying appropriate mechanisms-based treatment strategies for patients (see Figure 2). A variety of biopsychosocial mechanisms have been reported in the literature and will be explored in this section. Pain caused by COVID-19 has thought to be related to the systemic inflammation caused by one’s response to the infection, neuropathic mechanisms, and or pain caused by the viral infection itself and its treatments [53]. The proposed mechanism involves neuroinflammation through entry into the central nervous system and binding to the neuropilin-1 receptor at the olfactory cavity or by disruption of the blood–brain barrier [53]. Another mechanism involves angiotensin-2 mediated infiltration of viral entrance into human cells [54,55]. This mechanism may induce increased inflammation with a reciprocal anti-inflammatory response that is disproportionate leading to immunosuppression. This has also been shown as a cause of chronic pain after COVID-19 infection [55]. There has also been discussion around involvement with the sympathetic nervous system and its overactivity [53].

New onset of chronic pain had a prevalence of 19.6% in patients who had COVID-19 compared with only 1.4% in control patients [53]. Patients with a higher BMI, increasing age, more underlying disorders, and those who presented with myalgias were more likely to develop persistent musculoskeletal pain [53]. But most frequently, those that developed chronic pain secondary to COVID-19 were the patients who required intensive care for their infection. Patients requiring ICU admission, including those who received mechanical ventilation, neuromuscular blockers, and steroids, experienced an increased risk for developing myopathies, polyneuropathy, and muscle atrophy after COVID-19 infection [53]. Patients who were placed in prone positioning to help improve ventilation had an increase in peripheral nerve injury, joint subluxation, and soft tissue damage. All of which could contribute to the development of persistent pain after COVID-19 infection [53]. Some patients also experienced neuropathic pain secondary to therapeutic agents used for the treatment of active COVID-19 infection [54].

Many chronic pain patients, especially those with fibromyalgia, experienced an increase in pain during the pandemic. Patients struggled with receiving treatment as well as procedures as hospitals were focused on prioritizing the care of those infected with COVID-19. This led to an increase in cancellations as well as delays in treatments and interventions [53]. The pandemic also caused a reduction in physical activity, as many people were afraid to leave their homes, which ultimately led to increased pain [53]. Patients were also dealing with increased stress and anxiety from the ongoing changes regarding their job status and social distancing leading to physiological pain.

COVID-19 has caused many challenges in healthcare since the beginning and continues to leave an impact. Due to inflammation caused by the infection, patients have continued to deal with the long-lasting effects of COVID-19. Patients with chronic pain are no exception to this and we will likely continue to see an increase in patients living with pain due to COVID-19.

## 6. Management of Chronic Pain in the COVID-19 Era

### 6.1. Assessment of Chronic Pain Patients and Impact of COVID-19 Pandemic

The COVID-19 pandemic catalyzed rapid change in the way patients were seen and evaluated. Telemedicine provided the ability for patients to be evaluated in a safely distanced environment, while limiting the exposure of COVID-19 for chronic pain physicians, healthcare providers, and patients [56]. Telemedicine is an interactive communication platform with audio–visual capabilities conducted in real-time [57]. Telemedicine has many benefits for patients by providing access to healthcare services and minimizing the need for travel for patients living in remote areas or those with limited mobility [58]. Nevertheless, pre-COVID-19 systematic reviews indicate a lack of sufficient evidence to support its efficacy [59].

For healthcare workers, providing care for patients on a telemedicine platform presents challenges on different fronts. For patients with multiple co-morbidities, the need for in person consultation and physical examination cannot be fully replicated by a web-based platform [60,61]. Patients with multiple co-morbidities may also be at the highest risk for severe illness should they be exposed to COVID-19. The inability to perform a thorough physical examination and assessment of non-verbal cues may potentially lead to less accurate diagnosis when compared to in person consultations [60,61]. Additionally, the potential for technical difficulties during the patient interview and the need to spend additional time troubleshooting these issues also presents an issue for healthcare providers and patients alike [60].

As the transition to a post-COVID-era continues, telemedicine remains an effective tool in the assessment and management of chronic pain patients. Though it does not provide all the benefits of a thorough in person consultation, it provides accessibility to those who may have not been able to receive treatment before. The implementation of hybrid care models may be able to optimize individual treatment plans and improve patient outcomes [60] (see Figure 3).

### 6.2. Unique Diagnostic Challenges for Chronic Pain Physicians from COVID-19

Patients who are affected by chronic pain syndromes present a unique challenge to healthcare workers. The definition of chronic pain includes both a sensory and an emotional component [62]. Due to COVID-19, pandemic patients experienced increased depression, social isolation, and anxiety. Those who experience chronic pain may have worsened symptoms stemming from disrupted treatment, delay in evaluation, or reduced access to care [63]. Surveys from multiple countries demonstrated an increased pain burden in those already affected by chronic pain [53]. During the pandemic, specific disease states, like fibromyalgia, demonstrated a notable rise in baseline pain symptoms [64].

Despite the rapid implementation of web-based healthcare, patients with increased anxiety or pain burden were found by survey to be less accepting of telemedicine [65]. One survey of patients found that those who had treatments cancelled or postponed reported increased pain along with increased psychological distress [66].

For many, the emotional impact on overall quality of life was significantly decreased by the COVID-19 pandemic and subsequent lockdowns. This presented a unique diagnostic challenge for chronic pain physicians because of the emotional component associated with chronic pain. Emotions such as depression, stress, or anxiety can alter how one perceives pain [63]. This can lead to a negative feedback loop, where increased emotional hardship further propagates pain perception [62,67,68]. The assessment of the emotional aspect of chronic pain is imperative for comprehensive pain management.

### 6.3. Interplay between Persistent Symptoms after COVID-19 and Other Chronic Pain Conditions

As previously mentioned, after a COVID-19 infection, 10–20% of patients will go on to develop persistent symptoms after the recovery of their acute illness, commonly known as Long COVID. A sizable portion of these patients will have new development of pain resulting in an impact on everyday functioning. Some musculoskeletal manifestations of Long COVID include new-onset fatigue, arthralgia, myalgia, chest pain, and idiopathic pain syndromes [69].

While much research is emerging about new-onset pain associated with Long COVID, there remains little data specifically concerning the interaction between Long COVID and patients who had pre-existing chronic pain conditions. While the evidence suggests that the psychosocial aspects of the pandemic exacerbated chronic pain conditions, exactly how persistent symptoms of Long COVID affects patients with pre-existing chronic pain conditions remains an emerging area of research. Patients with pre-existing chronic pain were already at risk of increased pain during the pandemic. The persisting symptoms of Long COVID and the associated functional limitation superimposed upon these patients can lead to exacerbations of pain in a comparable manner with reduced access and delays to care.

It has been suggested that viral illnesses can lead to an exacerbation of previous chronic pain and serve as a trigger for new-onset chronic pain [60,70,71]. Pre-existing musculoskeletal pain has been shown to be significantly associated with musculoskeletal post-COVID pain even at 1 year after discharge [72]. In a single center study, most patients (67%) with chronic pain reported a persistent worsening of their previous chronic pain after a COVID-19 infection. A comparison of patients with new onset pain after COVID with those who had a history of chronic pain unrelated to COVID-19 showed that COVID-19 related new onset pain resulted in higher mean pain intensity, significantly more neuropathic pain symptoms, worse levels of anxiety and depression, worse cognitive function, and a greater effect on quality of life. Interestingly, the existence of previous chronic pain seemed to be protective against the development of a new chronic pain after COVID-19 infection [73].

### 6.4. Multidisciplinary Approach to Management of Chronic Pain in COVID-19

The prevalence of new-onset post-COVID-19 musculoskeletal pain was 73.2% [74]. This increased risk was likely due to a combination of cognitive, physical, and psychological factors brought on by the COVID-19 pandemic [75]. Patients with chronic pain are more likely to experience anxiety, depression, and poor sleep quality, which can all contribute to a lower quality of life [75]. Studies have shown that 65% of patients with depression complain of one or more loci of pain [76]. Chronic pain and depression have been demonstrated to activate intersecting brain structures, such as the thalamus, prefrontal cortex, and anterior cingulate cortex [77]. Additionally, the social isolation experienced by many during the pandemic worsened the experience of chronic pain and contributed to the development of depressive symptoms [78]. Accessibility to pain management services was also greatly diminished, exacerbating the dilemma of those suffering from chronic pain [79]. Due to the closure of many clinics and transition to telemedicine, patients were unable to access the same level of multidisciplinary care. In a sample of users of prescribed or over the counter pain medicines during the pandemic, 38% reported changes to their existing regimen, whereas 68% of patients reported changes to their non-pharmacologic pain management approaches [80]. The most common reasons for increased necessity of medication included changes in pain symptoms due to the pandemic, increased doses to compensate for lack of other physical or psychosocial interventions, or lack of access to providers [80]. 

Historically, it has been shown that a coordinated multidisciplinary approach that promotes physical and psychological interventions is most effective in treating pain [81]. Multidisciplinary pain regimens generally consist of cognitive behavioral therapy, psychoeducation, guided meditation, exercise therapy, dietitians, and finally, pharmacologic interventions. Patients should therefore receive proper education on neurophysiology, physical activity, and lifestyle modifications [82]. In COVID intensive care unit patients, a study performed by Ojeda et al. advocated for early care with therapeutic education and psychological intervention [83]. Research has shown that multiple types of exercise including yoga, aimed at improving coordination provide moderate pain benefits and minor functional benefits [84]. Cognitive behavioral therapy involving the use of breathing techniques and mindfulness training for stress reduction has also been effective in providing pain relief in fibromyalgia [84]. A literature review by Bair et al. examined the use of antidepressant therapy, and found a majority of studies analyzing SSRIs demonstrated improvement in both pain and depressive symptoms [76]. Consistent participation in exercise, physical therapy, and cognitive behavioral therapy have yielded the most consistently beneficial results in chronic pain reduction.

Opioids and non-steroidal anti-inflammatory drugs are some of the most used medications in the treatment of acute and chronic pain. Recently, a meta-analysis reported that COVID-19 patients that were utilizing opioids had an increase in overall mortality (OR 1.72, 95% CI 1.09–2.72, *p* = 0.02) and ICU admissions [85]. These results suggest that the use of opioids has potentially detrimental effects on COVID-19 prognosis through multiple mechanisms including immune modulation and respiratory depression [84]. The effects of opioids on the immune system are not well characterized but have been linked to infections [75]. Although pharmacologic interventions still have a place in the management of COVID-19-related pain syndromes, but special care must be taken with the recent trends in opioid misuse and safety concerns.

### 6.5. Emerging Techniques in Pain Management for COVID-19

Pain related to COVID-19 can be broadly divided into two categories, acute symptoms which are associated with the acute illness and the surrounding time-period and chronic symptoms that persist or appear after a COVID-19 infection and are thought to be related, often referred to as Long COVID symptoms. Acute symptoms most commonly include sore throat, myalgias, headaches, and arthralgias [86]. Non-pain-related symptoms that are often addressed at the same time as pain symptoms utilizing interventional techniques include loss of taste (ageusia), smell (anosmia), and fatigue [86,87].

Opioids and benzodiazepines may be used for pain control during acute COVID-19 infection [88]. In terms of sore throat and cough, opioids are often the mainstay for severe symptoms, though no particular opioid has been found to be superior [89]. Nonpharmacologic interventions such as psychological support, controlled breathing techniques, acupressure, music therapy, physical therapy, distraction, relaxation, and traditional Chinese medicine have also been employed with low levels of evidence [90]. Virtual reality systems have been successfully used to address a variety of needs in healthcare including successful treatment of pain in the inpatient setting which can translate to the management of COVID-19 pain [91,92].

In a study of 126 subjects, Hetherington et al. reported pharmacological treatment for symptom control as “effective” in 99 of 126 (79%) participants, “partially effective” in 24 of 126 (19%) participants, and “not effective” in 3 of 126 (2%) participants [93]. Strang et al. published results from a registry with a cohort of 390 subjects and demonstrated that 162 of 210 (77%) participants with pain had complete relief. Furthermore, 47 of 210 (22%) participants had partial relief of pain [94].

Fatigue, orthostatic intolerance, brain fog, anosmia, and ageusia/dysgeusia in Long COVID resemble the autonomic nervous system response to pro-inflammatory cytokines [95]. It is postulated that blocking cervical sympathetic chain activity with a local anesthetic allows the regional autonomic nervous system to reboot [96]. The dysregulated neuroinflammatory state observed in Long COVID-19 is thought to represent a subtle form of dysautonomia responsive to stellate ganglion block [97]. Liu and Duricka hypothesize that many of the symptoms of Long COVID present in a manner similar to CRPS but affect the brain rather than a limb [98]. Therefore, stellate ganglion blocks are effective in attenuating chronic sympathetic hyperresponsiveness, improving cerebral and regional blood flow, and recalibrating the autonomic nervous system toward pre-COVID homeostasis. The length of benefit is not tied to the duration of action of the local anesthetic because the block allows an opportunity for neuroimmune system reorganization or autoregulation which far outlasts the duration of the stellate ganglion block [98,99,100].

There are numerous case reports and case series in the literature that support the use of stellate ganglion block for management and treatment of Long COVID symptoms including pain [96,101,102,103]. Additional treatment protocols have been described in the literature incorporating a combination of targets. A 54-year-old gentleman with a constellation of Long COVID symptoms including pain underwent injections of 0.5% procaine in the stellate ganglion, sphenopalatine ganglion, and in clinically relevant points in the scalp, thorax, and abdomen three times over 3 months resulted in complete resolution of his symptoms [104]. Given the evolving nature of our understanding of Long COVID-19 and the application and mechanism of interventional treatments, it is inevitable that treatment algorithms will continue to evolve.

Several months into the response to COVID-19, Zhou et al. recognized that patients were experiencing respiratory compromise and extra ordinarily high mortality from excessive release of inflammatory mediators, known as the cytokine storm [105]. Thus, developing strategies that modulate the inflammatory response was thought to be a valuable tool in the battle against COVID. The vagus nerve is believed to control inflammation as well as bronchodilation in airway reactivity [106]. Staats et al. hypothesized that vagus nerve stimulation could be an adjunct to treatment in patients with COVID for the efferently mediated bronchodilation and the known ability to suppress inflammatory cytokines. Others later also suggested targeting the cholinergic anti-inflammatory pathway, via vagus nerve stimulation, to modulate the inflammatory response of COVID-19 [107]. In July of 2020, an emergency use authorization was granted by the FDA for use in patients with difficulty breathing in COVID-19, presumed to be due to exacerbation of asthma, making it one of the first medical devices approved to treat a symptom of COVID-19 [108]. Tornero et al. from Spain later demonstrated in a randomized controlled trial in patients admitted to the hospital with severe COVID-19 that noninvasive cervical vagus nerve stimulation could mitigate the inflammatory response in severe disease [109].

Verbank et al. found that in a subset of twenty patients with long COVID, mean duration of symptoms 22 weeks, that intensity of symptoms was decreased, and fatigue was significantly improved [110]. Badran et al. studied thirteen patients with auricular vagus nerve stimulation and noted that by 8 weeks those in the active group reported a 31% reduction in symptoms [111]. Ganesh, from Mayo Clinic is currently performing a randomized controlled trial of twenty patients looking at response of nVNS on Spike protein, cytokine panel, clinical symptoms scales and PET scanning [112]. Modulation of the vagus nerve continues to be a promising target for chronic pain associated with COVID-19.

## 7. Novel Pharmacological Treatments

As discussed in previous sections, multiple mechanisms for the development of COVID-19-related chronic pain symptoms have been proposed. Understanding the pathophysiology of this phenomenon can no doubt lead to clinicians potentially using mechanistically targeted pharmacotherapies to treat pain and other pain-related symptoms attributed to COVID-19 infection. To review, several hypotheses have been proposed, including immune activation, endothelial damage and microthrombosis, neuroinflammation due to microglial activation, and brain dysfunction due to the presence of angiotensin converting enzyme-2 and neuropilin-1 receptors [69,113]. It has been proposed that pain and pain-related symptoms due to COVID-19 infection are likely due to a combination of peripheral, central, and psychological components; however, further research is needed to fully elucidate the cause of such symptoms [114].

To date, there is no pharmacotherapy that is approved by the Food and Drug Administration (FDA) to treat COVID-19 chronic pain and pain-related symptoms. Several drug classes have been proposed as potential therapies including antihistamines and anti-depressants, although, additional novel pharmacotherapies aimed at other mechanisms of COVID-19-related pain can be considered [115]. To date, there are no specific guidelines regarding the topic of pharmacotherapy for the treatment of pain related to COVID-19 infection. Recently, Fernandez-de-las-Penas et al. suggested that to develop guidelines to treat post-COVID-19 chronic pain, we must first seek to understand the heterogenous presentation of Long COVID chronic pain and that through identification of the types of pain phenotypes (nociceptive, neuropathic, and nociplastic), we may move the needle towards developing precision pain guidelines [116]. The authors of this manuscript also stated that their literature search revealed no randomized, controlled, clinical trials evaluating the effectiveness of any particular treatment for post-COVID-19 chronic pain.

Nociceptive pain has been identified in COVID-19 chronic pain patients, but evidence supporting any particular treatment for the management of nociceptive pain in chronic COVID-19 pain is lacking. Post-COVID joint pain has been treated with nonsteroidal anti-inflammatory drugs (NSAIDs) and local steroids with potentially fair results [117], yet it should be mentioned that cardiovascular or cerebrovascular comorbidities of COVID-19 patients preclude the use of NSAIDs. Similarly, management of post-COVID-19 muscle pain has not received a significant amount of attention with well-designed clinical trials. A recent review of Cochrane evidence [118] and in a single patient case report [119], authors have highlighted the potential utility of physical therapy to treat post-COVID-19 related muscle and joint pain. Palmitoylethanolamide, an endocannabinoid-like lipid mediator with documented anti-inflammatory, analgesic, antimicrobial, immunomodulatory and neuroprotective effects, has also been proposed as a treatment for nociceptive pain [120]. It has been published that palmitoylethanolamide is a direct and indirect antiviral agent against SARS-CoV-2. Nevertheless, evidence supporting its use in COVID-19 is limited and only based on a case series with no discussion of its utility for post-COVID-19 pain [121,122].

Neuropathic pain has been more widely reported in COVID-19 chronic pain patients as painful peripheral neuropathies are often present in COVID-19 patients. Numerous agents and interventions have been recommended for the treatment of neuropathic pain in general, including but not limited to tricyclic antidepressants, serotonin–norepinephrine reuptake inhibitors, and gabapentinoids [123]. To-date, there are no studies to support any particular treatment, including those listed here, for the treatment of post-COVID-19 neuropathic pain.

As nociplastic pain is generally associated with widespread pain and associated symptoms such as fatigue, poor sleep, cognitive difficulties, and mood alterations, a multimodal and interdisciplinary approach is recommended to improve the possibility of successful treatment outcomes. As with post-COVID-19 related chronic nociceptive and neuropathic pain, there are no controlled trials to identify efficacy of treatments for nociplastic pain after COVID-19 infection. A recent pre-post single cohort intervention study has identified that low-dose naltrexone (LDN) has shown promising results in reducing symptoms of Long COVID-19 including pain [114]. In this study, a total of fifty-two participants were enrolled and a total of thirty-six participants finished the 2-month study where they initiated LDN at 1mg in month one and 2mg in month two. The outcome measures of interest included recovery from COVID-19, limitation of activities of daily living, energy levels, pain levels, levels of concentration and sleep disturbance as well as mood. After 2 months of LDN, statistically significant improvement in all measures except for mood were noted in this cohort, suggesting that LDN is safe and effective in reducing post-COVID-19 related symptoms including pain. 

In summary, although a great deal of literature exists to support varying pharmacologic and non-pharmacologic therapies for chronic nociplastic pain, neuropathic pain, and nociplastic pain of varying etiologies, there is a paucity of literature that exists to support specific pharmacologic agents in the treatment of these pain types in patients with post-COVID-19-related chronic pain. Further well-designed and rigorous research is indicated to identify mechanism-based and effective precision pharmacologic pain management strategies in this population.

## 8. Conclusions

The COVID-19 pandemic has caused many challenges in healthcare and continues to leave an impact, especially on patients suffering from chronic pain. There is still much to be learned regarding the mechanism by which COVID-19 causes chronic pain or interfaces with preexisting chronic pain syndromes. There is also a need for further study of emerging and novel approaches for the treatment of chronic pain associated with COVID-19 and to evolve evidence-based treatment options to mitigate the burden of suffering for the individual and its societal impact.

## Figures and Tables

**Figure 1 diagnostics-14-00332-f001:**
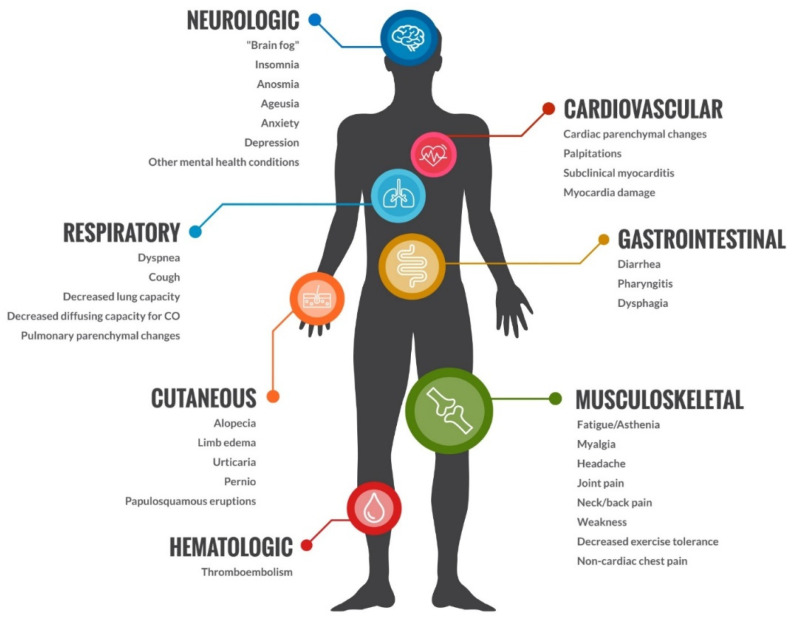
Effects of COVID on the body.

**Figure 2 diagnostics-14-00332-f002:**
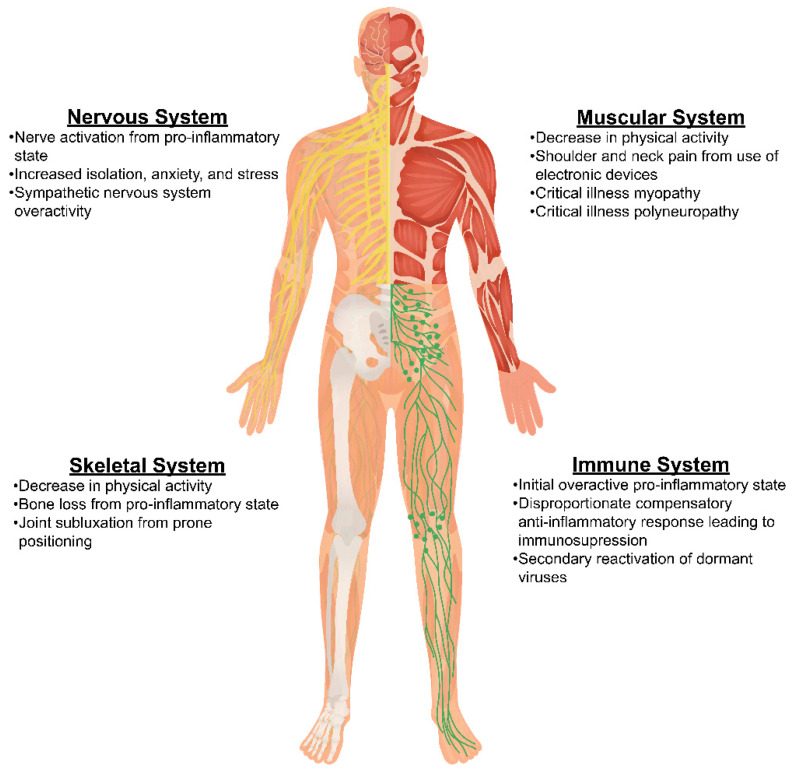
Mechanisms of COVID-19 on body systems.

**Figure 3 diagnostics-14-00332-f003:**
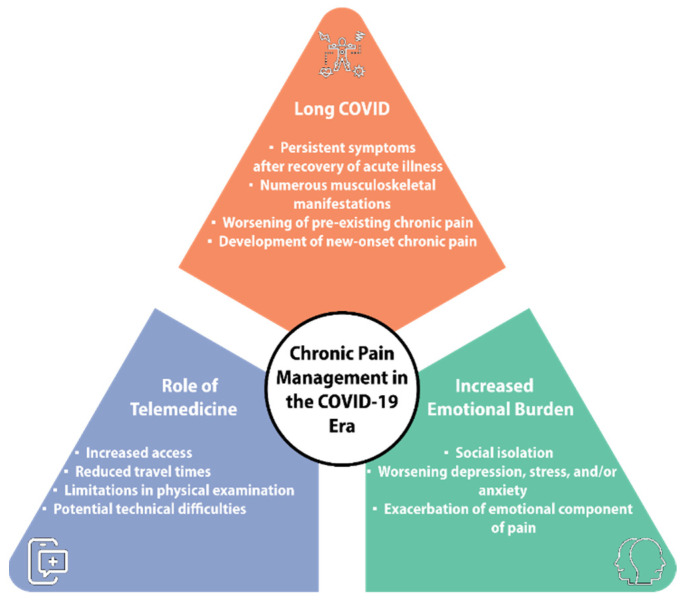
Chronic pain management in the COVID-19 Era.

## Data Availability

Not applicable.

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
