# Peer review of "Musculoskeletal and Neuropathic Pain in COVID-19"

_diagnostics, 2024, doi:10.3390/diagnostics14030332_

Round 1

Reviewer 1 Report

Comments and Suggestions for Authors

This is a very well written and interesting review about an important topic. However, I have some suggestions and comments which may improve the quality of this manuscript:

1.     Introduction

·  It provides a good overview of the issue of COVID-19 related chronic pain and the challenges it presents. However, the opening sentence is very broad

· chronic pain constitutes a burden globally in many disease states, not just COVID-19. Consider revising to more specifically introduce COVID-19 related chronic pain.

·  There are some lengthy sentences that could be shortened for clarity (e.g. lines 27-29). Breaking down complex ideas into shorter sentences can help with overall flow and readability.

·  Additional detail on some points would make the statements stronger. For example, what is the evidence that a multidisciplinary approach is most effective? A supporting reference would help back this claim.

·  The proposed novel techniques in lines 31-32 require more explanation as written now they seem out of place without proper context.

·  The conclusion calling for additional investigation is quite broad. Are there specific mechanisms of pain or management options that need further research? Being more targeted with the suggested research needs would strengthen the conclusion.

2.    Epidemiology of Musculoskeletal and Neuropathic Pain Pre-pandemic

·       The introduction focuses heavily on musculoskeletal rather than neuropathic pain. Consider balancing background more between both or clarifying purpose is to provide more detail on musculoskeletal as background.

·       Some lengthy sentences could be shortened or split for improved readability (e.g. lines 73-79).

·       The conclusion is abrupt, could benefit from summary sentence tying together key points on mechanisms and burden of these pain conditions.

·       Neuropathic pain section would be strengthened by adding more detail/examples on specific disease mechanisms that lead to neuropathic pain.

·       Some statements lack specific supporting references (e.g. line 59 on kinesiophobia as risk factor). Add citations where needed.

·       Format references in consistent style (e.g. numbers vs author name & year) across the whole manuscript.

3.     The Impact of COVID-19 on Healthcare

·       The introduction could provide more context on why analyzing the impacts on healthcare systems matters before diving into the details.

·       Some lengthy statistics-heavy sentences could be broken up or presented visually for increased readability.

·       The conclusion on telemedicine is abrupt - consider tying it back to the central theme of COVID-19 impacts rather than leaving it as an ancillary point.

·       There are opportunities to support statements more with citations e.g. quantifications of increased patient volumes, healthcare worker migration.

·       Could go into more detail on longer term impacts projected given health expenditures covered are within the first 5 months.

4.    COVID-19 and its Impact on Musculoskeletal and Neuropathic Pain

 · The introduction could relate back more explicitly to the previous sections on chronic pain and COVID-19's impact on healthcare instead of assuming reader knows context.

· Some statements about specific prevalence percentages and imaging findings lack supporting references.

· More detail explaining the mechanisms and theories behind neuropathic pain development post-COVID could strengthen that section.

5.    Emerging Perspectives on Underlying Mechanisms in Chronic Pain in COVID-19

·  Section is looking at chronic pain specifically associated with COVID-19 infection. Consider a bridge sentence making that transition for reader clarity.

·  A few lengthy, dense sentences could be broken up with supporting sub-bullets for readability (e.g. ICU admission risk factors).

·  The conclusion brings up stress/anxiety contributing to physiological pain but this point seems disconnected from earlier focus on COVID direct/treatment effects.

·  A summary sentence tying the mechanisms together would give conclusion more closure.

6.     Management of Chronic Pain in the COVID-19 Era

·There is some abrupt jumping between topics without transition sentences to orient reader to the new sections. Consider bridges between parts.

·A number of lengthy statistics-heavy sentences could be broken up for easier readability.

·The section on multidisciplinary care approach, while strong content, disrupts flow from previous diagnostic discussion into emerging techniques.

7.    Novel Pharmacological Treatments

·The introduction provides helpful background but could be more explicitly tied to the focus on pharmacological management options.

            ·Some sentences starting with multiple "however" statements sound   repetitive, consider rephrasing.

·Conclusion brings up need for precision medicine but this concept was not introduced earlier, so seems slightly out of place currently.

·Providing summary table of current evidence gaps could be helpful for reader.

Author Response

Please see attachment and thank you for your helpful comments and suggestions

Reviewer 2 Report

Comments and Suggestions for Authors

The paper is well written an interesting.

I would like to give you only minor suggestions:

I would be interesting to add a paragraph  and also a figure better explaining pathogenesis and interplay of different mechanism of pain in long COVID  (hyper inflammation, viral load or reactivation, gut microbiota disruption, and so on), maybe shorting a little bit the other paragraphs. 

Moreover, it would add more clarity a table reassuming pharmacological and non pharmacological intervention in long COVID pain as well as in acute pain during COVID, interventional treatments and so on. 

Round 2

Reviewer 1 Report

Comments and Suggestions for Authors

The authors responded to my comments very well. Thank you.